# Variational Information Maximisation for Intrinsically Motivated Reinforcement Learning

**Shakir Mohamed and Danilo J. Rezende**
Google DeepMind, London
{shakir, danilor}@google.com

## Abstract

The mutual information is a core statistical quantity that has applications in all areas of machine learning, whether this is in training of density models over multiple data modalities, in maximising the efficiency of noisy transmission channels, or when learning behaviour policies for exploration by artificial agents. Most learning algorithms that involve optimisation of the mutual information rely on the Blahut-Arimoto algorithm — an enumerative algorithm with exponential complexity that is not suitable for modern machine learning applications. This paper provides a new approach for scalable optimisation of the mutual information by merging techniques from variational inference and deep learning. We develop our approach by focusing on the problem of intrinsically-motivated learning, where the mutual information forms the definition of a well-known internal drive known as empowerment. Using a variational lower bound on the mutual information, combined with convolutional networks for handling visual input streams, we develop a stochastic optimisation algorithm that allows for scalable information maximisation and empowerment-based reasoning directly from pixels to actions.

## 1 Introduction

The problem of measuring and harnessing dependence between random variables is an inescapable statistical problem that forms the basis of a large number of applications in machine learning, including rate distortion theory [4], information bottleneck methods [28], population coding [1], curiosity-driven exploration [26, 21], model selection [3], and intrinsically-motivated reinforcement learning [22]. In all these problems the core quantity that must be reasoned about is the mutual information. In general, the mutual information (MI) is intractable to compute and few existing algorithms are useful for realistic applications. The received algorithm for estimating mutual information is the Blahut-Arimoto algorithm [31] that effectively solves for the MI by enumeration — an approach with exponential complexity that is not suitable for modern machine learning applications. By combining the best current practice from variational inference with that of deep learning, we bring the generality and scalability seen in other problem domains to information maximisation problems. We provide a new approach for maximisation of the mutual information that has significantly lower complexity, allows for computation with high-dimensional sensory inputs, and that allows us to exploit modern computational resources.

The technique we derive is generally applicable, but we shall describe and develop our approach by focussing on one popular and increasingly topical application of the mutual information: as a measure of 'empowerment' in intrinsically-motivated reinforcement learning. Reinforcement learning (RL) has seen a number of successes in recent years that has now established it as a practical, scalable solution for realistic agent-based planning and decision making [16, 13]. A limitation of the standard RL approach is that an agent is only able to learn using external rewards obtained from its environment; truly autonomous agents will often exist in environments that lack such external rewards or in environments where rewards are sparsely distributed. *Intrinsically-motivated reinforcement learning* [25] attempts to address this shortcoming by equipping an agent with a number of internal drives or intrinsic reward signals, such as hunger, boredom or curiosity that allows the agent to continue to explore, learn and act meaningfully in a reward-sparse world. There are many

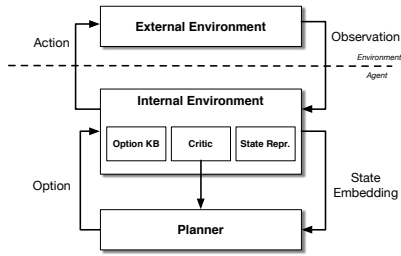

Figure 1: Perception-action loop separating environment into internal and external facets.

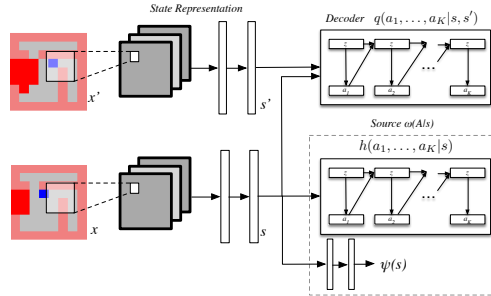

Figure 2: Computational graph for variational information maximisation.

ways in which to formally define internal drives, but what all such definitions have in common is that they, in some unsupervised fashion, allow an agent to reason about the value of information in the action-observation sequences it experiences. The mutual information allows for exactly this type of reasoning and forms the basis of one popular intrinsic reward measure, known as *empowerment*.

Our paper begins by describing the framework we use for online and self-motivated learning (section 2) and then describes the general problem associated with mutual information estimation and empowerment (section 3). We then make the following contributions:

- We develop *stochastic variational information maximisation*, a new algorithm for scalable estimation of the mutual information and channel capacity that is applicable to both discrete and continuous settings.
- We combine variational information optimisation and tools from deep learning to develop a *scalable algorithm for intrinsically-motivated reinforcement learning*, demonstrating a new application of the variational theory for problems in reinforcement learning and decision making.
- We demonstrate that empowerment-based behaviours obtained using variational information maximisation match those using the exact computation. We then apply our algorithms to a broad range of high-dimensional problems for which it is not possible to compute the exact solution, but for which we are able to act according to empowerment – *learning directly from pixel information*.

## 2 Intrinsically-motivated Reinforcement Learning

Intrinsically- or self-motivated learning attempts to address the question of where rewards come from and how they are used by an autonomous agent. Consider an online learning system that must model and reason about its incoming data streams and interact with its environment. This perception-action loop is common to many areas such as active learning, process control, black-box optimisation, and reinforcement learning. An extended view of this framework was presented by Singh et al. [25], who describe the environment as factored into external and internal components (figure 1). An agent receives observations and takes actions in the external environment. Importantly, the source and nature of any reward signals are not assumed to be provided by an oracle in the external environment, but is moved to an internal environment that is part of the agent's decision-making system; the internal environment handles the efficient processing of all input data and the choice and computation of an appropriate internal reward signal.

There are two important components of this framework: the state representation and the critic. We are principally interested in vision-based self-motivated systems, for which there are no solutions currently developed. To achieve this, our *state representation* system is a convolutional neural network [14]. The *critic* in figure 1 is responsible for providing intrinsic rewards that allow the agent to act under different types of internal motivations, and is where information maximisation enters the intrinsically-motivated learning problem.

The nature of the critic and in particular, the reward signal it provides is the main focus of this paper. A wide variety of reward functions have been proposed, and include: missing information or Bayesian surprise, which uses the KL divergence to measure the change in an agents internal belief after the observation of new data [8, 24]; measures based on prediction errors of future states such predicted $L_1$ change, predicted mode change or probability gain [17], or salient event prediction [25]; and measures based on information-theoretic quantities such as predicted information gain (PIG) [15], causal entropic forces [30] or empowerment [23]. The paper by Oudeyer & Kaplan [19]

currently provides the widest singular discussion of the breadth of intrinsic motivation measures. Although we have a wide choice of intrinsic reward measures, none of the available information-theoretic approaches are efficient to compute or scalable to high-dimensional problems: they require either knowledge of the true transition probability or summation over all configurations of the state space, which is not tractable for complex environments or when the states are large images.

## 3   Mutual Information and Empowerment

The mutual information is a core information-theoretic quantity that acts as a general measure of dependence between two random variables $\mathbf{x}$ and $\mathbf{y}$, defined as:

$$\mathcal{I}(\mathbf{x}, \mathbf{y}) = \mathbb{E}_{p(y|x)p(x)} \left[ \log \left( \frac{p(\mathbf{x}, \mathbf{y})}{p(\mathbf{x})p(\mathbf{y})} \right) \right], \tag{1}$$

where the $p(\mathbf{x}, \mathbf{y})$ is a joint distribution over the random variables, and $p(\mathbf{x})$ and $p(\mathbf{y})$ are the corresponding marginal distributions. $\mathbf{x}$ and $\mathbf{y}$ can be many quantities of interest: in computational neuroscience they are the sensory inputs and the spiking population code; in telecommunications they are the input signal to a channel and the received transmission; when learning exploration policies in RL, they are the current state and the action at some time in the future, respectively.

For intrinsic motivation, we use an internal reward measure referred to as empowerment [12, 23] that is obtained by searching for the maximal mutual information $\mathcal{I}(\cdot, \cdot)$, conditioned on a starting state $\mathbf{s}$, between a sequence of $K$ actions $\mathbf{a}$ and the final state reached $\mathbf{s}'$:

$$\mathcal{E}(\mathbf{s}) = \max_{\omega} \mathcal{I}^{\omega}(\mathbf{a}, \mathbf{s}'|\mathbf{s}) = \max_{\omega} \mathbb{E}_{p(s'|a,s)\omega(a|s)} \left[ \log \left( \frac{p(\mathbf{a}, \mathbf{s}'|\mathbf{s})}{\omega(\mathbf{a}|\mathbf{s})p(\mathbf{s}'|\mathbf{s})} \right) \right], \tag{2}$$

where $\mathbf{a} = \{a_1, \ldots, a_K\}$ is a sequence of $K$ primitive actions $a_k$ leading to a final state $\mathbf{s}'$, and $p(\mathbf{s}'|\mathbf{a}, \mathbf{s})$ is the $K$-step transition probability of the environment. $p(\mathbf{a}, \mathbf{s}'|\mathbf{s})$ is the joint distribution of action sequences and the final state, $\omega(\mathbf{a}|\mathbf{s})$ is a distribution over $K$-step action sequences, and $p(\mathbf{s}'|\mathbf{s})$ is the joint probability marginalised over the action sequence.

Equation (2) is the definition of the *channel capacity* in information theory and is a measure of the amount of information contained in the action sequences $\mathbf{a}$ about the future state $\mathbf{s}'$. This measure is compelling since it provides a well-grounded, task-independent measure for intrinsic motivation that fits naturally within the framework for intrinsically motivated learning described by figure 1. Furthermore, empowerment, like the state- or action-value function in reinforcement learning, assigns a value $\mathcal{E}(\mathbf{s})$ to each state $\mathbf{s}$ in an environment. An agent that seeks to maximise this value will move towards states from which it can reach the largest number of future states within its planning horizon $K$. It is this intuition that has led authors to describe empowerment as a measure of agent 'preparedness', or as a means by which an agent may quantify the extent to which it can reliably influence its environment — motivating an agent to move to states of maximum influence [23].

An empowerment-based agent generates an open-loop sequence of actions $K$ steps into the future — this is only used by the agent for its internal planning using $\omega(\mathbf{a}|\mathbf{s})$. When optimised using (2), the distribution $\omega(\mathbf{a}|\mathbf{s})$ becomes an efficient exploration policy that allows for uniform exploration of the state space reachable at horizon $K$, and is another compelling aspect of empowerment (we provide more intuition for this in appendix A). But this policy is not what is used by the agent for acting: when an agent must act in the world, it follows a closed-loop policy obtained by a planning algorithm using the empowerment value (e.g., Q-learning); we expand on this in sect. 4.3. A further consequence is that while acting, the agent is only 'curious' about parts of its environment that can be reached within its internal planning horizon $K$. We shall not explore the effect of the horizon in this work, but this has been widely-explored and we defer to the insights of Salge et al. [23].

## 4   Scalable Information Maximisation

The mutual information (MI) as we have described it thus far, whether it be for problems in empowerment, channel capacity or rate distortion, hides two difficult statistical problems. Firstly, computing the MI involves expectations over the unknown state transition probability. This can be seen by rewriting the MI in terms of the difference between conditional entropies $H(\cdot)$ as:

$$\mathcal{I}(\mathbf{a}, \mathbf{s}'|\mathbf{s}) = H(\mathbf{a}|\mathbf{s}) - H(\mathbf{a}|\mathbf{s}', \mathbf{s}), \tag{3}$$

where $H(\mathbf{a}|\mathbf{s}) = -\mathbb{E}_{\omega(a|s)}[\log \omega(\mathbf{a}|\mathbf{s})]$ and $H(\mathbf{a}|\mathbf{s}', \mathbf{s}) = -\mathbb{E}_{p(s'|a,s)\omega(a|s)}[\log p(\mathbf{a}|\mathbf{s}', \mathbf{s})]$. This computation requires marginalisation over the $K$-step transition dynamics of the environment $p(\mathbf{s}'|\mathbf{a}, \mathbf{s})$,

which is unknown in general. We could estimate this distribution by building a generative model of the environment, and then use this model to compute the MI. Since learning accurate generative models remains a challenging task, a solution that avoids this is preferred (and we also describe one approach for model-based empowerment in appendix B).

Secondly, we currently lack an efficient algorithm for MI computation. There exists no scalable algorithm for computing the mutual information that allows us to apply empowerment to high-dimensional problems and that allow us to easily exploit modern computing systems. The current solution is to use the Blahut-Arimoto algorithm [31], which essentially enumerates over all states, thus being limited to small-scale problems and not being applicable to the continuous domain. More scalable non-parametric estimators have been developed [7, 6]: these have a high memory footprint or require a very large number of observations, any approximation may not be a bound on the MI making reasoning about correctness harder, and they cannot easily be composed with existing (gradient-based) systems that allow us to design a unified (end-to-end) system. In the continuous domain, Monte Carlo integration has been proposed [10], but applications of Monte Carlo estimators can require a large number of draws to obtain accurate solutions and manageable variance. We have also explored Monte Carlo estimators for empowerment and describe an alternative importance sampling-based estimator for the MI and channel capacity in appendix B.1.

### 4.1 Variational Information Lower Bound

The MI can be made more tractable by deriving a lower bound to it and maximising this instead — here we present the bound derived by Barber & Agakov [1]. Using the entropy formulation of the MI (3) reveals that bounding the conditional entropy component is sufficient to bound the entire mutual information. By using the non-negativity property of the KL divergence, we obtain the bound:

$$\mathrm{KL}[p(x|y)\|q(x|y)] \geq 0 \Rightarrow H(x|y) \leq -\mathbb{E}_{p(x|y)}\left[\log q_\xi(x|y)\right]$$

$$\mathcal{I}^\omega(\mathbf{s}) = H(\mathbf{a}|\mathbf{s}) - H(\mathbf{a}|\mathbf{s}',\mathbf{s}) \geq H(\mathbf{a}) + \mathbb{E}_{p(s'|a,s)\omega_\theta(a|s)}[\log q_\xi(\mathbf{a}|\mathbf{s}',\mathbf{s})] = \mathcal{I}^{\omega,q}(\mathbf{s}) \qquad (4)$$

where we have introduced a variational distribution $q_\xi(\cdot)$ with parameters $\xi$; the distribution $\omega_\theta(\cdot)$ has parameters $\theta$. This bound becomes exact when $q_\xi(\mathbf{a}|\mathbf{s}',\mathbf{s})$ is equal to the true action posterior distribution $p(\mathbf{a}|\mathbf{s}',\mathbf{s})$. Other lower bounds for the mutual information are also possible: Jaakkola & Jordan [9] present a lower bound by using the convexity bound for the logarithm; Brunel & Nadal [2] use a Gaussian assumption and appeal to the Cramer-Rao lower bound.

The bound (4) is highly convenient (especially when compared to other bounds) since the transition probability $p(\mathbf{s}'|\mathbf{a},\mathbf{s})$ appears linearly in the expectation and we never need to evaluate its probability — we can thus evaluate the expectation directly by Monte Carlo using data obtained by interaction with the environment. The bound is also intuitive since we operate using the marginal distribution on action sequences $\omega_\theta(\mathbf{a}|\mathbf{s})$, which acts as a *source* (exploration distribution), the transition distribution $p(\mathbf{s}'|\mathbf{a},\mathbf{s})$ acts as an *encoder* (transition distribution) from $\mathbf{a}$ to $\mathbf{s}'$, and the variational distribution $q_\xi(\mathbf{a}|\mathbf{s}',\mathbf{s})$ conveniently acts as a *decoder* (planning distribution) taking us from $\mathbf{s}'$ to $\mathbf{a}$.

### 4.2 Variational Information Maximisation

A straightforward optimisation procedure based on (4) is an alternating optimisation for the parameters of the distributions $q_\xi(\cdot)$ and $\omega_\theta(\cdot)$. Barber & Agakov [1] made the connection between this approach and the generalised EM algorithm and refer to it as the IM (information maximisation) algorithm and we follow the same optimisation principle. From an optimisation perspective, the maximisation of the bound $\mathcal{I}^{\omega,q}(\mathbf{s})$ in (4) w.r.t. $\omega(\mathbf{a}|\mathbf{s})$ can be ill-posed (e.g., in Gaussian models, the variances can diverge). We avoid such divergent solutions by adding a constraint on the value of the entropy $H(\mathbf{a})$, which results in the constrained optimisation problem:

$$\hat{\mathcal{E}}(\mathbf{s}) = \max_{\omega,q} \mathcal{I}^{\omega,q}(\mathbf{s}) \; s.t. \; H(\mathbf{a}|\mathbf{s}) < \epsilon, \; \hat{\mathcal{E}}(\mathbf{s}) = \max_{\omega,q} \mathbb{E}_{p(s'|a,s)\omega(a|s)}[-\tfrac{1}{\beta}\ln\omega(\mathbf{a}|\mathbf{s}) + \ln q_\xi(\mathbf{a}|\mathbf{s}',\mathbf{s})] \quad (5)$$

where $\mathbf{a}$ is the action sequence performed by the agent when moving from $\mathbf{s}$ to $\mathbf{s}'$ and $\beta$ is an inverse temperature (which is a function of the constraint $\epsilon$).

At all times we use very general source and decoder distributions formed by complex non-linear functions using deep networks, and use stochastic gradient ascent for optimisation. We refer to our approach as *stochastic variational information maximisation* to highlight that we do all our computation on a mini-batch of recent experience from the agent. The optimisation for the decoder $q_\xi(\cdot)$ becomes a maximum likelihood problem, and the optimisation for the source $\omega_\theta(\cdot)$ requires computation of an unnormalised energy-based model, which we describe next. We summmarise the overall procedure in algorithm 1.

### 4.2.1 Maximum Likelihood Decoder

The first step of the alternating optimisation is the optimisation of equation (5) w.r.t. the decoder $q$, and is a supervised maximum likelihood problem. Given a set of data from past interactions with the environment, we learn a distribution from the start and termination states $\mathbf{s}, \mathbf{s}'$, respectively, to the action sequences $\mathbf{a}$ that have been taken. We parameterise the decoder as an auto-regressive distribution over the $K$-step action sequence:

$$q_\xi(\mathbf{a}|\mathbf{s}', \mathbf{s}) = q(a_1|\mathbf{s}, \mathbf{s}') \prod_{k=2}^{K} q(a_k | f_\xi(a_{k-1}, \mathbf{s}, \mathbf{s}')), \tag{6}$$

We are free to choose the distributions $q(a_k)$ for each action in the sequence, which we choose as categorical distributions whose mean parameters are the result of the function $f_\xi(\cdot)$ with parameters $\xi$. $f$ is a non-linear function that we specify using a two-layer neural network with rectified-linear activation functions. By maximising this log-likelihood, we are able to make stochastic updates to the variational parameters $\xi$ of this distribution. The neural network models used are expanded upon in appendix D.

### 4.2.2 Estimating the Source Distribution

Given a current estimate of the decoder $q$, the variational solution for the distribution $\omega(\mathbf{a}|\mathbf{s})$ computed by solving the functional derivative $\delta\mathcal{I}^\omega(s)/\delta\omega(\mathbf{a}|\mathbf{s}) = 0$ under the constraint that $\sum_a \omega(\mathbf{a}|\mathbf{s}) = 1$, is given by $\omega^\star(\mathbf{a}|\mathbf{s}) = \frac{1}{Z(s)} \exp\left(\hat{u}(\mathbf{s}, \mathbf{a})\right)$, where $u(\mathbf{s}, \mathbf{a}) = \mathbb{E}_{p(s'|s,a)}[\ln q_\xi(\mathbf{a}|\mathbf{s}, \mathbf{s}')]$, $\hat{u}(\mathbf{s}, \mathbf{a}) = \beta u(\mathbf{s}, \mathbf{a})$ and $Z(\mathbf{s}) = \sum_a e^{\hat{u}(s,a)}$ is a normalisation term. By substituting this optimal distribution into the original objective (5) we find that it can be expressed in terms of the normalisation function $Z(\mathbf{s})$ only, $\mathcal{E}(s) = \frac{1}{\beta}\log Z(\mathbf{s})$.

The distribution $\omega^\star(\mathbf{a}|\mathbf{s})$ is implicitly defined as an unnormalised distribution — there are no direct mechanisms for sampling actions or computing the normalising function $Z(\mathbf{s})$ for such distributions. We could use Gibbs or importance sampling, but these solutions are not satisfactory as they would require several evaluations of the unknown function $u(\mathbf{s}, \mathbf{a})$ per decision per state. We obtain a more convenient problem by approximating the unnormalised distribution $\omega^\star(\mathbf{a}|\mathbf{s})$ by a normalised (directed) distribution $h_\theta(\mathbf{a}|\mathbf{s})$. This is equivalent to approximating the energy term $\hat{u}(\mathbf{s}, \mathbf{a})$ by a function of the log-likelihood of the directed model, $r_\theta$:

$$\omega^\star(\mathbf{a}|\mathbf{s}) \approx h_\theta(\mathbf{a}|\mathbf{s}) \Rightarrow \hat{u}(\mathbf{s}, \mathbf{a}) \approx r_\theta(\mathbf{s}, \mathbf{a}); \qquad r_\theta(\mathbf{s}, \mathbf{a}) = \ln h_\theta(\mathbf{a}|\mathbf{s}) + \psi_\theta(\mathbf{s}). \tag{7}$$

We introduced a scalar function $\psi_\theta(\mathbf{s})$ into the approximation, but since this is not dependent on the action sequence $\mathbf{a}$ it does not change the approximation (7), and can be verified by substituting (7) into $\omega^\star(\mathbf{a}|\mathbf{s})$. Since $h_\theta(\mathbf{a}|\mathbf{s})$ is a normalised distribution, this leaves $\psi_\theta(\mathbf{s})$ to account for the normalisation term $\log Z(\mathbf{s})$, verified by substituting $\omega^\star(\mathbf{a}|\mathbf{s})$ and (7) into (5). We therefore obtain a cheap estimator of empowerment $\mathcal{E}(\mathbf{s}) \approx \frac{1}{\beta}\psi_\theta(\mathbf{s})$.

To optimise the parameters $\theta$ of the directed model $h_\theta$ and the scalar function $\psi_\theta$ we can minimise any measure of discrepancy between the two sides of the approximation (7). We minimise the squared error, giving the loss function $L(h_\theta, \psi_\theta)$ for optimisation as:

$$L(h_\theta, \psi_\theta) = \mathbb{E}_{p(s'|s,A)}\left[(\beta \ln q_\xi(\mathbf{a}|\mathbf{s}, \mathbf{s}') - r_\theta(\mathbf{s}, \mathbf{a}))^2\right]. \tag{8}$$

At convergence of the optimisation, we obtain a compact function with which to compute the empowerment that only requires forward evaluation of the function $\psi$. $h_\theta(\mathbf{a}|\mathbf{s})$ is parameterised using an auto-regressive distribution similar to (18), with conditional distributions specified by deep networks. The scalar function $\psi_\theta$ is also parameterised using a deep network. Further details of these networks are provided in appendix D.

### 4.3 Empowerment-based Behaviour policies

Using empowerment as an intrinsic reward measure, an agent will seek out states of maximal empowerment. We can treat the empowerment value $\mathcal{E}(\mathbf{s})$ as a state-dependent reward and can then utilise any standard planning algorithm, e.g., Q-learning, policy gradients or Monte Carlo search. We use the simplest planning strategy by using a one-step greedy empowerment maximisation. This amounts to choosing actions $a = \arg\max_a \mathcal{C}(\mathbf{s}, a)$, where $\mathcal{C}(\mathbf{s}, a) = \mathbb{E}_{p(s'|s,a)}[\mathcal{E}(\mathbf{s})]$. This policy does not account for the effect of actions beyond the planning horizon $K$. A natural enhancement is to use value iteration [27] to allow the agent to take actions by maximising its long term (potentially

Algorithm 1: Stochastic Variational Information
Maximisation for Empowerment
________________________________________

Parameters: $\xi$ variational, $\lambda$ convolutional, $\theta$ source

**while** not converged **do**

  $\mathbf{x} \leftarrow$ {Read current state}

  $\mathbf{s} = \mathrm{ConvNet}_\lambda(\mathbf{x})$ {Compute state repr.}

  $A \sim \omega(\mathbf{a}|\mathbf{s})$ {Draw action sequence.}

  Obtain data $(\mathbf{x}, \mathbf{a}, \mathbf{x}')$ {Acting in env. }

  $\mathbf{s}' = \mathrm{ConvNet}_\lambda(\mathbf{x}')$ {Compute state repr.}

  $\Delta\xi \propto \nabla_\xi \log q_\xi(\mathbf{a}|\mathbf{s},\mathbf{s}')$ (18)

  $\Delta\theta \propto \nabla_\theta L(h_\theta, \psi_\theta)$ (8)

  $\Delta\lambda \propto \nabla_\lambda \log q_\xi(\mathbf{a}|\mathbf{s},\mathbf{s}') + \nabla_\lambda L(h_\theta, \psi_\theta)$

**end while**

$\mathcal{E}(\mathbf{s}) = \frac{1}{\beta}\psi_\theta(\mathbf{s})$          {Empowerment}
________________________________________

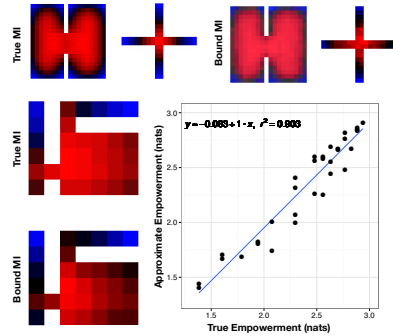

Figure 3: Comparing exact vs approximate empowerment. Heat maps: empowerment in 3 environments: two rooms, cross room, two-rooms; Scatter plot: agreement for two-rooms.

discounted) empowerment. A third approach would be to use empowerment as a potential function and the difference between the current and previous state's empowerment as a shaping function with in the planning [18]. A fourth approach is one where the agent uses the source distribution $\omega(\mathbf{a}|\mathbf{s})$ as its behaviour policy. The source distribution has similar properties to the greedy behaviour policy and can also be used, but since it effectively acts as an empowered agents internal exploration mechanism, it has a large variance (it is designed to allow uniform exploration of the state space). Understanding this choice of behaviour policy is an important line of ongoing research.

### 4.4 Algorithm Summary and Complexity

The system we have described is a scalable and general purpose algorithm for mutual information maximisation and we summarise the core components using the computational graph in figure 2 and in algorithm 1. The state representation mechanism used throughout is obtained by transforming raw observations $\mathbf{x}, \mathbf{x}'$ to produce the start and final states $\mathbf{s}, \mathbf{s}'$, respectively. When the raw observations are pixels from vision, the state representation is a *convolutional neural network* [14, 16], while for other observations (such as continuous measurements) we use a fully-connected neural network – we indicate the parameters of these models using $\lambda$. Since we use a unified loss function, we can apply gradient descent and backpropagate stochastic gradients through the entire model allowing for joint optimisation of both the information and representation parameters. For optimisation we use a preconditioned optimisation algorithm such as Adagrad [5].

The computational complexity of empowerment estimators involves the planning horizon $K$, the number of actions $N$, and the number of states $S$. For the exact computation we must enumerate over the number of states, which for grid-worlds is $S \propto D^2$ (for $D \times D$ grids), or for binary images is $S = 2^{D^2}$. The complexity of using the Blahut-Arimoto (BA) algorithm is $O(N^K S^2) = O(N^K D^4)$ for grid worlds or $O(N^K 2^{2D^2})$ for binary images. The BA algorithm, even in environments with a small number of interacting objects becomes quickly intractable, since the state space grows exponentially with the number of possible interactions, and is also exponential in the planning horizon. In contrast, our approach deals directly on the image dimensions. Using visual inputs, the convolutional network produces a vector of size $P$, upon which all subsequent computation is based, consisting of an $L$-layer neural network. This gives a complexity for state representation of $O(D^2 P + LP^2)$. The autoregressive distributions have complexity of $O(H^2 KN)$, where $H$ is the size of the hidden layer. Thus, our approach has at most quadratic complexity in the size of the hidden layers used and linear in other quantities, and matches the complexity of any currently employed large-scale vision-based models. In addition, since we use gradient descent throughout, we are able to leverage the power of GPUs and distributed gradient computations.

## 5 Results

We demonstrate the use of empowerment and the effectiveness of variational information maximisation in two types of environments. *Static environments* consists of rooms and mazes in different configurations in which there are no objects with which the agent can interact, or other moving ob-



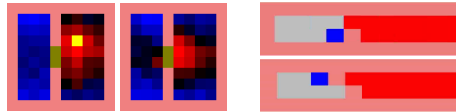

Figure 4: Empowerment for a room environment, showing a) an empty room, b) room with an obstacle c) room with a moveable box, d) room with row of moveable boxes.

Figure 5: Left: empowerment landscape for agent and key scenario. Yellow is the key and green is the door. Right: Agent in a corridor with flowing lava. The agent places a bricks to stem the flow of lava.

jects. The number of states in these settings is equal to the number of locations in the environment, so is still manageable for approaches that rely on state enumeration. In *dynamic environments*, aspects of the environment change, such as flowing lava that causes the agent to reset, or a predator that chases the agent. For the most part, we consider discrete action settings in which the agent has five actions (up, down, left, right, do nothing). The agent may have other actions, such as picking up a key or laying down a brick. There are no external rewards available and the agent must reason purely using visual (pixel) information. For all these experiments we used a horizon of $K = 5$.

## 5.1 Effectiveness of the MI Bound

We first establish that the use of the variational information lower bound results in the same behaviour as that obtained using the *exact* mutual information in a set of static environments. We consider environments that have at most 400 discrete states and compute the true mutual information using the Blahut-Arimoto algorithm. We compute the variational information bound on the same environment using pixel information (on $20 \times 20$ images). To compare the two approaches we look at the empowerment landscape obtained by computing the empowerment at every location in the environment and show these as heatmaps. For action selection, what matters is the location of the maximum empowerment, and by comparing the heatmaps in figure 3, we see that the empowerment landscape matches between the exact and the variational solution, and hence, will lead to the same agent-behaviour.

In each image in figure 3, we show a heat-map of the empowerment for each location in the environment. We then analyze the point of highest empowerment: for the large room it is in the centre of the room; for the cross-shaped room it is at the centre of the cross, and in a two-rooms environment, it is located near both doors. In addition, we show that the empowerment values obtained by our method constitute a close approximation to the true empowerment for the two-rooms environment (correlation coeff = 1.00, $R^2$=0.90). These results match those by authors such as Klyubin et al. [12] (using empowerment) and Wissner-Gross & Freer [30] (using a different information-theoretic measure — the causal entropic force). The advantage of the variational approach is clear from this discussion: we are able to obtain solutions of the same quality as the exact computation, we have far more favourable computational scaling (one that is not exponential in the size of the state space and planning horizon), and we are able to plan directly from pixel information.

## 5.2 Dynamic Environments

Having established the usefulness of the bound and some further understanding of empowerment, we now examine the empowerment behaviour in environments with dynamic characteristics. Even in small environments, the number of states becomes extremely large if there are objects that can be moved, or added and removed from the environment, making enumerative algorithms (such as BA) quickly infeasible, since we have an exponential explosion in the number of states. We first reproduce an experiment from Salge et al. [23, §4.5.3] that considers the empowered behaviour of an agent in a room-environment, a room that: is empty, has a fixed box, has a moveable box, has a row of moveable boxes. Salge et al. [23] explore this setup to discuss the choice of the state representation, and that not including the existence of the box severely limits the planning ability of the agent. In our approach, we do not face this problem of choosing the state representation, since the agent will reason about all objects that appear within its visual observations, obviating the need for hand-designed state representations. Figure 4 shows that in an empty room, the empowerment is uniform almost everywhere except close to the walls; in a room with a fixed box, the fixed box limits the set of future reachable states, and as expected, empowerment is low around the box; in a room where the box can be moved, the box can now be seen as a tool and we have high empowerment near the box; similarly, when we have four boxes in a row, the empowerment is highest around the

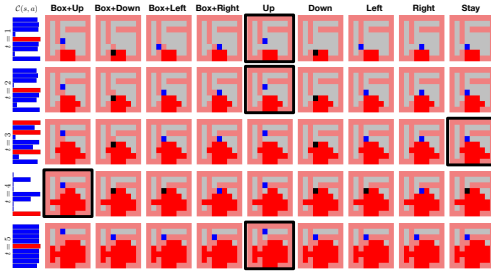

Figure 6: Empowerment planning in a lava-filled maze environment. Black panels show the path taken by the agent.

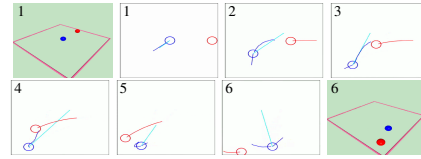

Figure 7: Predator (red) and agent (blue) scenario. Panels 1, 6 show the 3D simulation. Other panels show a trace of the path that the predator and prey take at points on its trajectory. The blue/red shows path history; cyan shows the direction to the maximum empowerment.

boxes. These results match those of Salge et al. [23] and show the effectiveness of reasoning from pixel information directly.

Figure 6 shows how planning with empowerment works in a dynamic maze environment, where lava flows from a source at the bottom that eventually engulfs the maze. The only way the agent is able to safeguard itself, is to stem the flow of lava by building a wall at the entrance to one of the corridors. At every point in time $t$, the agent decides its next action by computing the expected empowerment after taking one action. In this environment, we show the planning for all 9 available actions and a bar graph with the empowerment values for each resulting state. The action that leads to the highest empowerment is taken and is indicated by the black panels[1].

Figure 5(left) shows two-rooms separated by a door. The agent is able to collect a key that allows it to open the door. Before collecting the key, the maximum empowerment is in the region around the key, once the agent has collected the key, the region of maximum empowerment is close to the door[2]. Figure 5(right) shows an agent in a corridor and must protect itself by building a wall of bricks, which it is able to do successfully using the same empowerment planning approach described for the maze setting.

### 5.3 Predator-Prey Scenario

We demonstrate the applicability of our approach to continuous settings, by studying a simple 3D physics simulation [29], shown in figure 7. Here, the agent (blue) is followed by a predator (red) and is randomly reset to a new location in the environment if caught by the predator. Both the agent and the predator are represented as spheres in the environment that roll on a surface with friction. The state is the position, velocity and angular momentum of the agent and the predator, and the action is a 2D force vector. As expected, the maximum empowerment lies in regions away from the predator, which results in the agent learning to escape the predator[3].

## 6 Conclusion

We have developed a new approach for scalable estimation of the mutual information by exploiting recent advances in deep learning and variational inference. We focussed specifically on intrinsic motivation with a reward measure known as empowerment, which requires at its core the efficient computation of the mutual information. By using a variational lower bound on the mutual information, we developed a scalable model and efficient algorithm that expands the applicability of empowerment to high-dimensional problems, with the complexity of our approach being extremely favourable when compared to the complexity of the Blahut-Arimoto algorithm that is currently the standard. The overall system does not require a generative model of the environment to be built, learns using only interactions with the environment, and allows the agent to learn directly from visual information or in continuous state-action spaces. While we chose to develop the algorithm in terms of intrinsic motivation, the mutual information has wide applications in other domains, all which stand to benefit from a scalable algorithm that allows them to exploit the abundance of data and be applied to large-scale problems.

**Acknowledgements:** We thank Daniel Polani for invaluable guidance and feedback.

## Footnotes

[1] Video: http://youtu.be/eA9jVDa7O38   [2] Video: http://youtu.be/eSAIJ0isc3Y   [3] Videos: http://youtu.be/tMiiKXPirAQ; http://youtu.be/LV5jYY-JFpE

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
