[Reviews · NeurIPS 2015]

Submitted by Assigned_Reviewer_1

*** This paper was added to my pile at the last minute ***. so all my evaluation is a guess. First I don't think I fully understand the importance of the objective. Since I am not very familiar with reinforcement learning and empowerment, I have some difficulty to understand the novel contribution.
Summary: This paper proposed some a new approach for maximizing mutual information.

Submitted by Assigned_Reviewer_2

This paper uses a neural net as a variational approximation to estimate the mutual information between current action and future state of a reinforcement learning agent, in order to estimate the so-called "empowerment" of a particular state. I'm not very familiar with the field of reinforcement learning, so it's hard to comment on the novelty and importance of a new way to estimate empowerment. However, it does seem that variational inference has already been used in this context, by Barber and Agakov, and that the main innovation in this work is just to use a neural net for the proposal distribution.

The paper is fairly well-written, although a little unclear in places, and the technical content seems sound. My biggest crticism is that the experiments are very weak. There are no baseline or comparison experiments at all. Surely the authors should compare their approach with simper (i.e. non-neural) variational approximations, and also with MCMC and other approaches. As far as I can tell, the entire conclusion of the experiments was "we tried it on some toy problems and it seemed to work ok".

Overall, a paper of possible interest to the RL community, but I'm not convinced that it will have a major impact at NIPS.
Summary: Possibly relevant to the RL community, but lacks convincing experiments and broad appeal.

Submitted by Assigned_Reviewer_3

The paper concerns a stochastic variational approach towards mutual information maximisation with applications to reinforcement learning. The authors present a treatment of MI following a bound taken form earlier work by Barber&Agakov and using it stochastically by mini-batch descent. In order to estimate the MI they introduce a novelty and use neural networks to predict parameters for factors in state transition models as shown in Equation 6. This replaces the need for an explicit generative model of the data. The authors use this algorithm in the context of empowerment, which is a measure that can be used to connect MI with reinforcement learning. they demonstrate very simple empowerment based policies using their MI approximation which yield experimentally good results while running much more efficiently than previous algorithms used to estimate the MI.

The paper is written clearly and appears to be of good quality throughout. The organization is slightly perplexing, as frequently the purpose of the paper is mixed up between the efficient estimation of MI and reinforcement learning and I

would urge the authors to clarify what their precise goals are: the paper is not sufficiently focused on reinforcement learning to be evaluated as such and it also fails to clearly show broad applicability for stochastic variational MI in non-reinforcement learning settings.

That said, the results appear encouraging and efficient flexible methods for modeling actions in environments using neural networks are currently relevant directions for research.

Comparisons with Auto-Encoding Variational Bayes would be interesting, as the approximation of factors using neural networks in a variational setting is reminiscent of this paper. I would also be interested in seeing how exactly

the performance of the model depends on the length of the action states given that the variational stochastic approximation to the gaussian factors does not use any kind of control variates to help learning of these factors and the fact that a factorization over many variational distributions will become worse and worse as the number of the factors grows without modeling interactions of components.
Summary: The authors present a stochastic algorithm for MI which is based on previous work and add the novelty of estimating factors given neural networks. The paper is confusingly organized in its topic, but otherwise well written and procures experimental support for its claims.

Author Feedback
Author rebuttal: We thank the reviewers for the discussion. Our paper lies at the intersection of approximate inference, deep learning and reinforcement learning, and the reviews highlight why this discussion is needed. We break our response into two parts to address the main themes and concerns.

-- Motivation and significance (R1,2,4)
The mutual information is a core information theoretic quantity with many useful features, though we rarely see it applied in machine learning. One of the principal reasons for this is the difficulty in computing and optimizing it. The existing algorithm (i.e. Blahut Arimoto) is a brute-force algorithm - something faster and more scalable is needed for modern machine learning, which is what we present in this paper. There are other approximations we highlight, but the variational lower bound of Barber and Agakov (2004) is in our view one that has the most potential to be scaled. The 'importance of this objective' [R1] is that with scalable methods we will finally see interesting applications of the MI, such as empowerment, information bottleneck, rate distortion, spatial coherence, etc.

It is not true that we do a 'straightforward combination of existing techniques' [R4]:
1. The initial work of Barber and Agakov shows that such an MI bound exists, however, they applied this to relatively simple and low-dimensional problems like Gaussians or Gaussians mixtures. For these cases the required expectations can be either solved analytically or easily approximated. This is not the case we present.
2. We extend greatly upon the work of Barber and Agakov [R1, 2,4]: by introducing the more stable (trust region) objective eq (5), by considering Monte Carlo evaluations thus avoiding the need to build a generative model, by using non-linear autoregressive distributions rather than Gaussians, developing a new optimisation scheme eqs (7)-(8) that simplifies later use of the model, by showing that such an estimation can be used with challenging colour-image data using deep learning, and by applying it for the first time in the reinforcement learning domain.
3. We do not 'just use neural networks for the proposal' [R2]. For neural networks to be useful for MI optimization several manipulations must be performed. This includes the introduction a variational bound, finding an effective parametrization of the variational policy q and the optimization scheme in eqs (7,8) which overcomes the need to learn an explicit transition model.

--Comparisons and alternatives (R2,3,4)

We chose to focus on one interesting and increasingly topical application of the mutual information, i.e. intrinsically-motivated reinforcement learning using empowerment. This is more interesting than demonstrations on simple Gaussian channels and exposes the problems of using existing MI solvers. This is of interest to our entire ML community as we strive to build more autonomous learning systems.

The statement 'no baseline or comparison experiments' [R2] is not true - we compare our scheme directly to the exact solution provided by the BA algorithm for a number of cases in fig 3, and is the approach taken in all existing work (e.g., [20]). The results indicate that our method provides a reliable approximation to the exact method. We then apply our method to problems where the state-space is too large for the exact method - recall BA is exponential in the size of the state space (sect 4.4).

-- Specific comments
R2: We parameterise our distributions to make them more powerful using deep/auto-regressive nets but simpler distributions can easily be used, and this is separate from the use of variational inference. We have also derived in the appendix a MC optimizer for completeness, but even this estimator is prohibitively expensive for most interesting problems.

R3: Your comments on the paper arrangement are appreciated and we will improve the presentation by separating the MI bound and its computation from its application to empowerment.
As you suggested, an alternative approach is to develop a generative model using VAEs (and was our initial inspiration), but constructing fully reliable generative models of images is still hard and something we wished to avoid at this point. The length of the horizon is an important topic and we wish to investigate it further.

R4: We describe the neural networks used in the appendix in more detail. We provide comparisons to the exact solution. We designed the parts of our algorithm to work together: the bound provides a unified loss function for stochastic optimisation; the convolutional network is what allows us to reason only in the image size rather than the state-space size and is what reduces complexity from exponential to quadratic; the inner optimisation (7)(8) generalises the empowerment computation and allows for fast use at test time.